# Beyond Pairwise: Provably Fast Algorithms for Approximate k-Way Similarity Search

**Anshumali Shrivastava**
Department of Computer Science
Computing and Information Science
Cornell University
Ithaca, NY 14853, USA
anshu@cs.cornell.edu

**Ping Li**
Department of Statistics & Biostatistics
Department of Computer Science
Rutgers University
Piscataway, NJ 08854, USA
pingli@stat.rutgers.edu

## Abstract

We go beyond the notion of pairwise similarity and look into search problems with $k$-way similarity functions. In this paper, we focus on problems related to **3-way Jaccard** similarity: $\mathcal{R}^{3way} = \frac{|S_1 \cap S_2 \cap S_3|}{|S_1 \cup S_2 \cup S_3|}$, $S_1, S_2, S_3 \in \mathcal{C}$, where $\mathcal{C}$ is a size $n$ collection of sets (or binary vectors). We show that approximate $\mathcal{R}^{3way}$ similarity search problems admit fast algorithms with provable guarantees, analogous to the pairwise case. Our analysis and speedup guarantees naturally extend to $k$-way resemblance. In the process, we extend traditional framework of *locality sensitive hashing (LSH)* to handle higher-order similarities, which could be of independent theoretical interest. The applicability of $\mathcal{R}^{3way}$ search is shown on the "Google Sets" application. In addition, we demonstrate the advantage of $\mathcal{R}^{3way}$ resemblance over the pairwise case in improving retrieval quality.

## 1 Introduction and Motivation

Similarity search (near neighbor search) is one of the fundamental problems in Computer Science. The task is to identify a small set of data points which are "most similar" to a given input query. Similarity search algorithms have been one of the basic building blocks in numerous applications including search, databases, learning, recommendation systems, computer vision, etc.

One widely used notion of similarity on sets is the Jaccard similarity or *resemblance* [5, 10, 18, 20]. Given two sets $S_1, S_2 \subseteq \Omega = \{0, 1, 2, ..., D-1\}$, the resemblance $\mathcal{R}^{2way}$ between $S_1$ and $S_2$ is defined as: $\mathcal{R}^{2way} = \frac{|S_1 \cap S_2|}{|S_1 \cup S_2|}$. Existing notions of similarity in search problems mainly work with pairwise similarity functions. In this paper, we go beyond this notion and look at the problem of $k$-way similarity search, where the similarity function of interest involves $k$ sets ($k \geq 2$). Our work exploits the fact that resemblance can be naturally extended to $k$-way resemblance similarity [18, 21], defined over $k$ sets $\{S_1, S_2, ..., S_k\}$ as $\mathcal{R}^{k-way} = \frac{|S_1 \cap S_2 \cap ... \cap S_k|}{|S_1 \cup S_2 \cup ... \cup S_k|}$.

**Binary high-dimensional data**    The current web datasets are typically binary, sparse, and extremely high-dimensional, largely due to the wide adoption of the "Bag of Words" (BoW) representations for documents and images. It is often the case, in BoW representations, that just the presence or absence (0/1) of specific feature words captures sufficient information [7, 16, 20], especially with (e.g.,) 3-grams or higher-order models. And so, the web can be imagined as a giant storehouse of ultra high-dimensional sparse binary vectors. Of course, binary vectors can also be equivalently viewed as sets (containing locations of the nonzero features).

We list **four** practical scenarios where $k$-way resemblance search would be a natural choice.

**(i) Google Sets:**    (http://googlesystem.blogspot.com/2012/11/google-sets-still-available.html) *Google Sets* is among the earliest google projects, which allows users to generate list of similar words by typing only few related keywords. For example, if the user types "mazda" and "honda" the application will automatically generate related words like "bmw", "ford", "toyota", etc. This application is currently available in google spreadsheet. If we assume the term document binary representation of each word $w$ in the database, then given query $w_1$ and $w_2$, we show that $\frac{|w_1 \cap w_2 \cap w|}{|w_1 \cup w_2 \cup w|}$ turns out to be a very good similarity measure for this application (see Section 7.1).

**(ii) Joint recommendations:** Users A and B would like to watch a movie together. The profile of each person can be represented as a sparse vector over a giant universe of attributes. For example, a user profile may be the set of actors, actresses, genres, directors, etc, which she/he likes. On the other hand, we can represent a movie M in the database over the same universe based on attributes associated with the movie. If we have to recommend movie M, jointly to users A and B, then a natural measure to maximize is $\frac{|A \cap B \cap M|}{|A \cup B \cup M|}$. The problem of group recommendation [3] is applicable in many more settings such as recommending people to join circles, etc.

**(iii) Improving retrieval quality:** We are interested in finding images of a particular type of object, and we have two or three (possibly noisy) representative images. In such a scenario, a natural expectation is that retrieving images simultaneously similar to all the representative images should be more refined than just retrieving images similar to any one of them. In Section 7.2, we demonstrate that in cases where we have more than one element to search for, we can refine our search quality using $k$-way resemblance search. In a dynamic feedback environment [4], we can improve subsequent search quality by using $k$-way similarity search on the pages already clicked by the user.

**(iv) Beyond pairwise clustering:** While machine learning algorithms often utilize the data through pairwise similarities (e.g., inner product or resemblance), there are natural scenarios where the affinity relations are not pairwise, but rather triadic, tetradic or higher [2, 30]. The computational cost, of course, will increase exponentially if we go beyond pairwise similarity.

**Efficiency is crucial** With the data explosion in modern applications, the brute force way of scanning all the data for searching is prohibitively expensive, specially in user-facing applications like search. The need for $k$-way similarity search can only be fulfilled if it admits efficient algorithms. This paper fulfills this requirement for $k$-way resemblance and its derived similarities. In particular, we show fast algorithms with provable query time guarantees for approximate $k$-way resemblance search. Our algorithms and analysis naturally provide a framework to extend classical LSH framework [14, 13] to handle higher-order similarities, which could be of independent theoretical interest.

**Organization** In Section 2, we review approximate near neighbor search and classical Locality Sensitive Hashing (LSH). In Section 3, we formulate the 3-way similarity search problems. Sections 4, 5, and 6 describe provable fast algorithms for several search problems. Section 7 demonstrates the applicability of 3-way resemblance search in real applications.

## 2 Classical $c$-NN and Locality Sensitive Hashing (LSH)

Initial attempts of finding efficient (sub-linear time) algorithms for exact near neighbor search, based on space partitioning, turned out to be a disappointment with the massive dimensionality of current datasets [11, 28]. Approximate versions of the problem were proposed [14, 13] to break the linear query time bottleneck. One widely adopted formalism is the $c$-approximate near neighbor ($c$-NN).

**Definition 1** (*c-Approximate Near Neighbor or c-NN*). *Consider a set of points, denoted by P, in a D-dimensional space $\mathbb{R}^D$, and parameters $R_0 > 0$, $\delta > 0$. The task is to construct a data structure which, given any query point q, if there exist an $R_0$-near neighbor of q in P, it reports some $cR_0$-near neighbor of q in P with probability $1 - \delta$.*

The usual notion of $c$-NN is for distance. Since we deal with similarities, we define $R_0$-near neighbor of point $q$ as a point $p$ with $Sim(q, p) \geq R_0$, where $Sim$ is the similarity function of interest.

*Locality sensitive hashing (LSH)* [14, 13] is a popular framework for $c$-NN problems. LSH is a family of functions, with the property that similar input objects in the domain of these functions have a higher probability of colliding in the range space than non-similar ones. In formal terms, consider $\mathcal{H}$ a family of hash functions mapping $\mathbb{R}^D$ to some set $\mathcal{S}$

**Definition 2** (*Locality Sensitive Hashing (LSH)*). *A family $\mathcal{H}$ is called $(R_0, cR_0, p_1, p_2)$-sensitive if for any two points $x, y \in \mathbb{R}^D$ and h chosen uniformly from $\mathcal{H}$ satisfies the following:*

- *if $Sim(x, y) \geq R_0$ then $Pr_{\mathcal{H}}(h(x) = h(y)) \geq p_1$*
- *if $Sim(x, y) \leq cR_0$ then $Pr_{\mathcal{H}}(h(x) = h(y)) \leq p_2$*

For approximate nearest neighbor search typically, $p_1 > p_2$ and $c < 1$ is needed. Note, $c < 1$ as we are defining neighbors in terms of similarity. Basically, LSH trades off query time with extra preprocessing time and space which can be accomplished off-line.

**Fact 1** *Given a family of $(R_0, cR_0, p_1, p_2)$ -sensitive hash functions, one can construct a data structure for c-NN with $O(n^\rho \log_{1/p_2} n)$ query time and space $O(n^{1+\rho})$, where $\rho = \frac{\log 1/p_1}{\log 1/p_2}$.*

**Minwise Hashing for Pairwise Resemblance**  One popular choice of LSH family of functions associated with resemblance similarity is, *Minwise Hashing family* [5, 6, 13]. Minwise Hashing family applies an independent random permutation $\pi : \Omega \to \Omega$, on the given set $S \subseteq \Omega$, and looks at the minimum element under $\pi$, i.e. $\min(\pi(S))$. Given two sets $S_1, S_2 \subseteq \Omega = \{0, 1, 2, ..., D-1\}$, it can be shown by elementary probability argument that

$$Pr\left(\min(\pi(S_1)) = \min(\pi(S_2)))\right) = \frac{|S_1 \cap S_2|}{|S_1 \cup S_2|} = \mathcal{R}^{2way}. \tag{1}$$

The recent work on $b$-bit minwise hashing [20, 23] provides an improvement by storing only the lowest $b$ bits of the hashed values: $\min(\pi(S_1)), \min(\pi(S_2))$. [26] implemented the idea of building hash tables for near neighbor search, by directly using the bits from $b$-bit minwise hashing.

## 3   3-way Similarity Search Formulation

Our focus will remain on binary vectors which can also be viewed as sets. We illustrate our method using 3-way resemblance similarity function $Sim(S_1, S_2, S_3) = \frac{|S_1 \cap S_2 \cap S_3|}{|S_1 \cup S_2 \cup S_3|}$. The algorithm and guarantees naturally extend to $k$-way resemblance. Given a size $n$ collection $\mathcal{C} \subseteq 2^\Omega$ of sets (or binary vectors), we are particularly interested in the following three problems:

1. Given two query sets $S_1$ and $S_2$, find $S_3 \in \mathcal{C}$ that maximizes $Sim(S_1, S_2, S_3)$.
2. Given a query set $S_1$, find two sets $S_2, S_3 \in \mathcal{C}$ maximizing $Sim(S_1, S_2, S_3)$.
3. Find three sets $S_1, S_2, S_3 \in \mathcal{C}$ maximizing $Sim(S_1, S_2, S_3)$.

The brute force way of enumerating all possibilities leads to the worst case query time of $O(n)$, $O(n^2)$ and $O(n^3)$ for problem 1, 2 and 3, respectively. In a hope to break this barrier, just like the case of pairwise near neighbor search, we define the $c$-approximate ($c < 1$) versions of the above three problems. As in the case of $c$-NN, we are given two parameters $R_0 > 0$ and $\delta > 0$. For each of the following three problems, the guarantee is with probability at least $1 - \delta$:

1. (**3-way $c$-Near Neighbor or 3-way $c$-NN**) Given two query sets $S_1$ and $S_2$, if there exists $S_3 \in \mathcal{C}$ with $Sim(S_1, S_2, S_3) \geq R_0$, then we report some $S_3' \in \mathcal{C}$ so that $Sim(S_1, S_2, S_3') \geq cR_0$.
2. (**3-way $c$-Close Pair or 3-way $c$-CP**) Given a query set $S_1$, if there exists a pair of set $S_2, S_3 \in \mathcal{C}$ with $Sim(S_1, S_2, S_3) \geq R_0$, then we report sets $S_2', S_3' \in \mathcal{C}$ so that $Sim(S_1, S_2', S_3') \geq cR_0$.
3. (**3-way $c$-Best Cluster or 3-way $c$-BC**) If there exist sets $S_1, S_2, S_3 \in \mathcal{C}$ with $Sim(S_1, S_2, S_3) \geq R_0$, then we report sets $S_1', S_2', S_3' \in \mathcal{C}$ so that $Sim(S_1', S_2', S_3') \geq cR_0$.

## 4   Sub-linear Algorithm for 3-way $c$-NN

The basic philosophy behind sub-linear search is bucketing, which allows us to preprocess dataset in a fashion so that we can filter many bad candidates without scanning all of them. LSH-based techniques rely on randomized hash functions to create buckets that probabilistically filter bad candidates. This philosophy is not restricted for binary similarity functions and is much more general. Here, we first focus on 3-way $c$-NN problem for binary data.

**Theorem 1** *For $\mathcal{R}^{3way}$ c-NN one can construct a data structure with $O(n^\rho \log_{1/cR_0} n)$ query time and $O(n^{1+\rho})$ space, where $\rho = 1 - \frac{\log 1/c}{\log 1/c + \log 1/R_0}$.*  □

The argument for 2-way resemblance can be naturally extended to $k$-way resemblance. Specifically, given three sets $S_1, S_2, S_3 \subseteq \Omega$ and an independent random permutation $\pi : \Omega \to \Omega$, we have:

$$Pr\left(\min(\pi(S_1)) = \min(\pi(S_2)) = \min(\pi(S_3)))\right) = \mathcal{R}^{3way}. \tag{2}$$

Eq.( 2) shows that minwise hashing, although it operates on sets individually, preserves all 3-way (in fact $k$-way) similarity structure of the data. The existence of such a hash function is the key requirement behind the existence of efficient approximate search. For the pairwise case, the probability event was a simple hash collision, and the min-hash itself serves as the bucket index. In case

of 3-way (and higher) $c$-NN problem, we have to take care of a more complicated event to create an indexing scheme. In particular, during preprocessing we need to create buckets for each individual $S_3$, and while querying we need to associate the query sets $S_1$ and $S_2$ to the appropriate bucket. We need extra mechanisms to manipulate these minwise hashes to obtain a bucketing scheme.

**Proof of Theorem 1:** We use two additional functions: $f_1 : \Omega \to N$ for manipulating $\min(\pi(S_3))$ and $f_2 : \Omega \times \Omega \to N$ for manipulating both $\min(\pi(S_1))$ and $\min(\pi(S_2))$. Let $a \in \mathbb{N}^+$ such that $|\Omega| = D < 10^a$. We define $f_1(x) = (10^a + 1) \times x$ and $f_2(x, y) = 10^a x + y$. This choice ensures that given query $S_1$ and $S_2$, for any $S_3 \in \mathcal{C}$, $f_1(\min(\pi(S_3))) = f_2(\min(\pi(S_1)), \min(\pi(S_2)))$ holds if and only if $\min(\pi(S_1)) = \min(\pi(S_2)) = \min(\pi(S_2))$, and thus we get a bucketing scheme. To complete the proof, we introduce two integer parameters $K$ and $L$. Define a new hash function by concatenating $K$ events. To be more precise, while preprocessing, for every element $S_3 \in \mathcal{C}$ create buckets $g_1(S_3) = [f_1(h_1(S_3)); ...; f_1(h_K(S_3))]$ where $h_i$ is chosen uniformly from minwise hashing family. For given query points $S_1$ and $S_2$, retrieve only points in the bucket $g_2(S_1, S_2) = [f_2(h_1(S_1), h_1(S_2)); ...; f_2(h_K(S_1), h_K(S_2))]$. Repeat this process $L$ times independently. For any $S_3 \in \mathcal{C}$, with $Sim(S_1, S_2, S_3) \geq R_0$, is retrieved with probability at least $1 - (1 - R_0^K)^L$. Using $K = \lceil \frac{\log n}{\log \frac{1}{cR_0}} \rceil$ and $L = \lceil n^\rho \log(\frac{1}{\delta}) \rceil$, where $\rho = 1 - \frac{\log 1/c}{\log 1/c + \log 1/R_0}$, the proof can be obtained using standard concentration arguments used to prove Fact 1, see [14, 13]. It is worth noting that the probability guarantee parameter $\delta$ gets absorbed in the constants as $\log(\frac{1}{\delta})$. Note, the process is stopped as soon as we find some element with $\mathcal{R}^{3way} \geq cR_0$. $\qquad\square$

Theorem 1 can be easily extended to $k$-way resemblance with same query time and space guarantees. Note that $k$-way $c$-NN is at least as hard as $k^*$-way c-NN for any $k^* \leq k$, because we can always choose $(k - k^* + 1)$ identical query sets in $k$-way $c$-NN, and it reduces to $k^*$-way $c$-NN problem. So, any improvements in $\mathcal{R}^{3way}$ $c$-NN implies improvement in the classical min-hash LSH for Jaccard similarity. The proposed analysis is thus tight in this sense.

The above observation makes it possible to also perform the traditional pairwise $c$-NN search using the same hash tables deployed for 3-way $c$-NN. In the query phase we have an option, if we have two different queries $S_1, S_2$, then we retrieve from bucket $g_2(S_1, S_2)$ and that is usual 3-way $c$-NN search. If we are just interested in pairwise near neighbor search given one query $S_1$, then we will look into bucket $g_2(S_1, S_1)$, and we know that the 3-way resemblance between $S_1, S_1, S_3$ boils down to the pairwise resemblance between $S_1$ and $S_3$. So, the same hash tables can be used for both the purposes. This property generalizes, and hash tables created for $k$-way $c$-NN can be used for any $k^*$-way similarity search so long as $k^* \leq k$. The approximation guarantees still holds. This flexibility makes $k$-way $c$-NN bucketing scheme more advantageous over the pairwise scheme.

One of the peculiarity of LSH based techniques is that the query complexity exponent $\rho < 1$ is dependent on the choice of the threshold $R_0$ we are interested in and the value of $c$ which is the approximation ratio that we will tolerate. Figure 1 plots $\rho = 1 - \frac{\log 1/c}{\log 1/c + \log 1/R_0}$ with respect to $c$, for selected $R_0$ values from 0.01 to 0.99. For instance, if we are interested in highly similar pairs, i.e. $R_0 \approx 1$, then we are looking at near $O(\log n)$ query complexity for $c$-NN problem as $\rho \approx 0$. On the other hand, for very lower threshold $R_0$, there is no much of hope of time-saving because $\rho$ is close to 1.

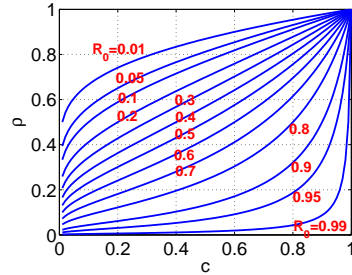

Figure 1: $\rho = 1 - \frac{\log 1/c}{\log 1/c + \log 1/R_0}$.

## 5  Other Efficient $k$-way Similarities

We refer to the $k$-way similarities for which there exist sub-linear algorithms for $c$-NN search with query and space complexity exactly as given in Theorem 1 as **efficient** . We have demonstrated existence of one such example of efficient similarities, which is the $k$-way resemblance. This leads to a natural question: "Are there more of them?".

[9] analyzed all the transformations on similarities that preserve existence of efficient LSH search. In particular, they showed that if $\mathcal{S}$ is a similarity for which there exists an LSH family, then there also exists an LSH family for any similarity which is a *probability generating function (PGF)* transformation on $\mathcal{S}$. PGF transformation on $\mathcal{S}$ is defined as $PGF(\mathcal{S}) = \sum_{i=1}^{\infty} p_i \mathcal{S}^i$, where $\mathcal{S} \in [0, 1]$ and $p_i \geq 0$ satisfies $\sum_{i=1}^{\infty} p_i = 1$. Similar theorem can also be shown in the case of 3-way resemblance.

**Theorem 2** *Any PGF transformation on* 3*-way resemblance* $\mathcal{R}^{3way}$ *is* efficient. □

Recall in the proof of Theorem 1, we created hash assignments $f_1(\min(\pi(S_3)))$ and $f_2(\min(\pi(S_1)), \min(\pi(S_2)))$, which lead to a bucketing scheme for the 3-way resemblance search, where the collision event $E = \{f_1(\min(\pi(S_3)) = f_2(\min(\pi(S_1)), \min(\pi(S_2)))\}$ happens with probability $Pr(E) = \mathcal{R}^{3way}$. To prove the above Theorem 2, we will need to create hash events having probability $PGF(\mathcal{R}^{3way}) = \sum_{i=1}^{\infty} p_i(\mathcal{R}^{3way})^i$. Note that $0 \leq PGF(\mathcal{R}^{3way}) \leq 1$. We will make use of the following simple lemma.

**Lemma 1** $(\mathcal{R}^{3way})^n$ *is* efficient *for all* $n \in \mathbb{N}$.

**Proof:** Define new hash assignments $g_1^n(S_3) = [f_1(h_1(S_3)); ...; f_1(h_n(S_3))]$ and $g_2^n(S_1, S_2) = [f_2(h_1(S_1), h_1(S_2)); ...; f_2(h_n(S_1), h_n(S_2))]$. The collision event $g_1^n(S_3) = g_2^n(S_1, S_2)$ has probability $(\mathcal{R}^{3way})^n$. We now use the pair $< g_1^n, g_2^n >$ instead of $< f_1, f_2 >$ and obtain same guarantees, as in Theorem 1, for $(\mathcal{R}^{3way})^n$ as well. □

**Proof of Theorem 2:** From Lemma 1, let $< g_1^i, g_2^i >$ be the hash pair corresponding to $(\mathcal{R}^{3way})^i$ as used in above lemma. We sample one hash pair from the set $\{< g_1^i, g_2^i >: i \in \mathbb{N}\}$, where the probability of sampling $< g_1^i, g_2^i >$ is proportional to $p_i$. Note that $p_i \geq 0$, and satisfies $\sum_{i=1}^{\infty} p_i = 1$, and so the above sampling is valid. It is not difficult to see that the collision of the sampled hash pair has probability exactly $\sum_{i=1}^{\infty} p_i(\mathcal{R}^{3way})^i$. □

Theorem 2 can be naturally extended to $k$-way similarity for any $k \geq 2$. Thus, we now have infinitely many $k$-way similarity functions admitting efficient sub-linear search. One, that might be interesting, because of its radial basis kernel like nature, is shown in the following corollary.

**Corollary 1** $e^{\mathcal{R}^{k-way}-1}$ *is* efficient.

**Proof:** Use the expansion of $e^{\mathcal{R}^{k-way}}$ normalized by e to see that $e^{\mathcal{R}^{k-way}-1}$ is a PGF on $\mathcal{R}^{k-way}$. □

## 6 Fast Algorithms for 3-way $c$-CP and 3-way $c$-BC Problems

For 3-way $c$-CP and 3-way $c$-BC problems, using bucketing scheme with minwise hashing family will save even more computations.

**Theorem 3** *For* $\mathcal{R}^{3way}$ *c-Close Pair Problem (or c-CP) one can construct a data structure with* $O(n^{2\rho} \log_{1/cR_0} n)$ *query time and* $O(n^{1+2\rho})$ *space, where* $\rho = 1 - \frac{\log 1/c}{\log 1/c + \log 1/R_0}$. □

Note that we can switch the role of $f_1$ and $f_2$ in the proof of Theorem 1. We are thus left with a $c$-NN problem with search space $O(n^2)$ (all pairs) instead of n. A bit of analysis, similar to Theorem 1, will show that this procedure achieves the required query time $O(n^{2\rho} \log_{1/cR_0} n)$, but uses a lot more space, $O(n^{2(1+\rho)})$, than shown in the above theorem. It turns out that there is a better way of doing $c$-CP that saves us space.

**Proof of Theorem 3:** We again start with constructing hash tables. For every element $S_c \in \mathcal{C}$, we create a hash-table and store $S_c$ in bucket $B(S_c) = [h_1(S_c); h_2(S_c); ...; h_K(S_c)]$, where $h_i$ is chosen uniformly from minwise independent family of hash functions $\mathcal{H}$. We create $L$ such hash-tables. For a query element $S_q$ we look for all pairs in bucket $B(S_q) = [h_1(S_q); h_2(S_q); ...; h_K(S_q)]$ and repeat this for each of the $L$ tables. Note, we do not form pairs of elements retrieved from different tables as they do not satisfy Eq. (2). If there exists a pair $S_1, S_2 \in \mathcal{C}$ with $Sim(S_q, S_1, S_2) \geq R_0$, using Eq. (2), we can see that we will find that pair in bucket $B(S_q)$ with probability $1 - (1 - R_0^K)^L$. Here, we cannot use traditional choice of $K$ and $L$, similar to what we did in Theorem 1, as there are $O(n^2)$ instead of $O(n)$ possible pairs. We instead use $K = \lceil \frac{2 \log n}{\log \frac{1}{cR_0}} \rceil$ and $L = \lceil n^{2\rho} \log(\frac{1}{\delta}) \rceil$, with $\rho = 1 - \frac{\log 1/c}{\log 1/c + \log 1/R_0}$. With this choice of $K$ and $L$, the result follows. Note, the process is stopped as soon as we find pairs $S_1$ and $S_2$ with $Sim(S_q, S_1, S_2) \geq cR_0$. The key argument that saves space from $O(n^{2(1+\rho)})$ to $O(n^{1+2\rho})$ is that we hash $n$ points individually. Eq. (2) makes it clear that hashing all possible pairs is not needed when every point can be processed individually, and pairs formed within each bucket itself filter out most of the unnecessary combinations. □

**Theorem 4** *For $\mathcal{R}^{3way}$ c-Best Cluster Problem (or c-BC) there exist an algorithm with running time $O(n^{1+2\rho} \log_{1/cR_0} n)$, where $\rho = 1 - \frac{\log 1/c}{\log 1/c + \log 1/R_0}$.* □

The argument similar to one used in proof of Theorem 3 leads to the running time of $O(n^{1+3\rho} \log_{1/cR_0} n)$ as we need $L = O(n^{3\rho})$, and we have to processes all points at least once.

**Proof of Theorem 4:** Repeat $c$-CP problem $n$ times for every element in collection $\mathcal{C}$ acting as query once. We use the same set of hash tables and hash functions every time. The preprocessing time is $O(n^{1+2\rho} \log_{1/cR_0} n)$ evaluations of hash functions and the total querying time is $O(n \times n^{2\rho} \log_{1/cR_0} n)$, which makes the total running time $O(n^{1+2\rho} \log_{1/cR_0} n)$. □

For $k$-way $c$-BC Problem, we can achieve $O(n^{1+(k-1)\rho} \log_{1/cR_0} n)$ running time. If we are interested in very high similarity cluster, with $R_0 \approx 1$, then $\rho \approx 0$, and the running time is around $O(n \log n)$. This is a huge saving over the brute force $O(n^k)$. In most practical cases, specially in big data regime where we have enormous amount of data, we can expect the $k$-way similarity of good clusters to be high and finding them should be efficient. We can see that with increasing $k$, hashing techniques save more computations.

## 7 Experiments

In this section, we demonstrate the usability of 3-way and higher-order similarity search using (i) Google Sets, and (ii) Improving retrieval quality.

### 7.1 Google Sets: Generating Semantically Similar Words

Here, the task is to retrieve words which are "semantically" similar to the given set of query words. We collected 1.2 million random documents from Wikipedia and created a standard term-doc binary vector representation of each term present in the collected documents after removing standard stop words and punctuation marks. More specifically, every word is represented as a 1.2 million dimension binary vector indicating its presence or absence in the corresponding document. The total number of terms (or words) was around 60,000 in this experiment.

Since there is no standard benchmark available for this task, we show qualitative evaluations. For querying, we used the following four pairs of semantically related words: (i) "jaguar" and "tiger"; (ii) "artificial" and "intelligence"; (iii) "milky" and "way" ; (iv) "finger" and "lakes". Given the query words $w_1$ and $w_2$, we compare the results obtained by the following four methods.

- **Google Sets:** We use Google's algorithm and report 5 words from Google spreadsheets [1]. This is Google's algorithm which uses its own data.
- **3-way Resemblance (3-way):** We use 3-way resemblance $\frac{|w_1 \cap w_2 \cap w|}{|w_1 \cup w_2 \cup w|}$ to rank every word $w$ and report top 5 words based on this ranking.
- **Sum Resemblance (SR):** Another intuitive method is to use the sum of pairwise resemblance $\frac{|w_1 \cap w|}{|w_1 \cup w|} + \frac{|w_2 \cap w|}{|w_2 \cup w|}$ and report top 5 words based on this ranking.
- **Pairwise Intersection (PI):** We first retrieve top 100 words based on pairwise resemblance for each $w_1$ and $w_2$ independently. We then report the words common in both. If there is no word in common we do not report anything.

The results in Table 1 demonstrate that using 3-way resemblance retrieves reasonable candidates for these four queries. An interesting query is "finger" and "lakes". Finger Lakes is a region in upstate New York. Google could only relate it to New York, while 3-way resemblance could even retrieve the names of cities and lakes in the region. Also, for query "milky" and "way", we can see some (perhaps) unrelated words like "dance" returned by Google. We do not see such random behavior with 3-way resemblance. Although we are not aware of the algorithm and the dataset used by Google, we can see that 3-way resemblance appears to be a right measure for this application.

The above results also illustrate the problem with using the sum of pairwise similarity method. The similarity value with one of the words dominates the sum and hence we see for queries "artificial" and "intelligence" that all the retrieved words are mostly related to the word "intelligence". Same is the case with query "finger" and "lakes" as well as "jaguar" and "tiger". Note that "jaguar" is also a car brand. In addition, for all 4 queries, there was no common word in the top 100 words similar to the each query word individually and so PI method never returns anything.

Table 1: Top five words retrieved using various methods for different queries.

| "JAGUAR" AND " TIGER" | | | | "ARTIFICIAL" AND "INTELLIGENCE" | | | |
|---|---|---|---|---|---|---|---|
| GOOGLE | 3-WAY | SR | PI | GOOGLE | 3-WAY | SR | PI |
| LION | LEOPARD | CAT | — | COMPUTER | COMPUTER | SECURITY | — |
| LEOPARD | CHEETAH | LEOPARD | — | PROGRAMMING | SCIENCE | WEAPONS | — |
| CHEETAH | LION | LITRE | — | SCIENCE | INTELLIGENT | SECRET | — |
| CAT | PANTHER | BMW | — | ROBOT | HUMAN | ATTACKS | — |
| DOG | CAT | CHASIS | — | ROBOTICS | TECHNOLOGY | HUMAN | — |

| "MILKY" AND " WAY" | | | | "FINGER" AND "LAKES" | | | |
|---|---|---|---|---|---|---|---|
| GOOGLE | 3-WAY | SR | PI | GOOGLE | 3-WAY | SR | PI |
| DANCE | GALAXY | EVEN | — | NEW | SENECA | RIVERS | — |
| STARS | STARS | ANOTHER | — | YORK | CAYUGA | FRESHWATER | — |
| SPACE | EARTH | STILL | — | NY | ERIE | FISH | — |
| THE | LIGHT | BACK | — | PARK | ROCHESTER | STREAMS | — |
| UNIVERSE | SPACE | TIME | — | CITY | IROQUOIS | FORESTED | — |

We should note the importance of the denominator term in 3-way resemblance, without which frequent words will be blindly favored. The exciting contribution of this paper is that 3-way resemblance similarity search admits provable sub-linear guarantees, making it an ideal choice. On the other hand, no such provable guarantees are known for SR and other heuristic based search methods.

### 7.2 Improving Retrieval Quality in Similarity Search

We also demonstrate how the retrieval quality of traditional similarity search can be boosted by utilizing more query candidates instead of just one. For the evaluations we choose two public datasets: MNIST and WEBSPAM, which were used in a recent related paper [26] for near neighbor search with binary data using $b$-bit minwise hashing [20, 23].

The two datasets reflect diversity both in terms of task and scale that is encountered in practice. The MNIST dataset consists of handwritten digit samples. Each sample is an image of $28 \times 28$ pixel yielding a 784 dimension vector with the associated class label (digit $0 - 9$). We binarize the data by settings all non zeros to be 1. We used the standard partition of MNIST, which consists of 10,000 samples in one set and 60,000 in the other. The WEBSPAM dataset, with 16,609,143 features, consists of sparse vector representation of emails labeled as spam or not. We randomly sample 70,000 data points and partitioned them into two independent sets of size 35,000 each.

Table 2: Percentage of top candidates with the same labels as that of query retrieved using various similarity criteria. More indicates better retrieval quality (Best marked in **bold**).

| TOP | MNIST | | | | WEBSPAM | | | |
|---|---|---|---|---|---|---|---|---|
| | 1 | 10 | 20 | 50 | 1 | 10 | 20 | 50 |
| *Pairwise* | 94.20 | 92.33 | 91.10 | 89.06 | 98.45 | 96.94 | 96.46 | 95.12 |
| *3-way NNbor* | 96.90 | 96.13 | 95.36 | 93.78 | 99.75 | 98.68 | 97.80 | 96.11 |
| *4-way NNbor* | **97.70** | **96.89** | **96.28** | **95.10** | **99.90** | **98.87** | **98.15** | **96.45** |

For evaluation, we need to generate potential similar search query candidates for $k$-way search. It makes no sense in trying to search for object simultaneously similar to two very different objects. To generate such query candidates, we took one independent set of the data and partition it according to the class labels. We then run a cheap k-mean clustering on each class, and randomly sample triplets $< x_1, x_2, x_3 >$ from each cluster for evaluating 2-way, 3-way and 4-way similarity search. For MNIST dataset, the standard 10,000 test set was partitioned according to the labels into 10 sets, each partition was then clustered into 10 clusters, and we choose 10 triplets randomly from each cluster. In all we had 100 such triplets for each class, and thus 1000 overall query triplets. For WEBSPAM, which consists only of 2 classes, we choose one of the independent set and performed the same procedure. We selected 100 triplets from each cluster. We thus have 1000 triplets from each class making the total number of 2000 query candidates.

The above procedures ensure that the elements in each triplets $< x_1, x_2, x_3 >$ are not very far from each other and are of the same class label. For each triplet $< x_1, x_2, x_3 >$, we sort all the points $x$ in the other independent set based on the following:

- **Pairwise:** We only use the information in $x_1$ and rank $x$ based on resemblance $\frac{|x_1 \cap x|}{|x_1 \cup x|}$.

- **3-way NN:** We rank $x$ based on 3-way resemblance $\frac{|x_1 \cap x_2 \cap x|}{|x_1 \cup x_2 \cup x|}$.
- **4-way NN:** We rank $x$ based on 4-way resemblance $\frac{|x_1 \cap x_2 \cap x_3 \cap x|}{|x_1 \cup x_2 \cup x_3 \cup x|}$.

We look at the top 1, 10, 20 and 50 points based on orderings described above. Since, all the query triplets are of the same label, The percentage of top retrieved candidates having same label as that of the query items is a natural metric to evaluate the retrieval quality. This percentage values accumulated over all the triplets are summarized in Table 2.

We can see that top candidates retrieved by 3-way resemblance similarity, using 2 query points, are of better quality than vanilla pairwise similarity search. Also 4-way resemblance, with 3 query points, further improves the results compared to 3-way resemblance similarity search. This clearly demonstrates that multi-way resemblance similarity search is more desirable whenever we have more than one representative query in mind. Note that, for MNIST, which contains 10 classes, the boost compared to pairwise retrieval is substantial. The results follow a consistent trend.

## 8   Future Work

While the work presented in this paper is promising for efficient 3-way and $k$-way similarity search in binary high-dimensional data, there are numerous interesting and practical research problems we can study as future work. In this section, we mention a few such examples.

***One-permutation hashing***. Traditionally, building hash tables for near neighbor search required many (e.g., 1000) independent hashes. This is both time- and energy-consuming, not only for building tables but also for processing un-seen queries which have not been processed. *One permutation hashing* [22] provides the hope of reducing many permutations to merely one. The version in [22], however, was not applicable to near neighbor search due to the existence of many empty bins (which offer no indexing capability). The most recent work [27] is able to fill the empty bins and works well for pairwise near neighbor search. It will be interesting to extend [27] to $k$-way search.

***Non-binary sparse data***. This paper focuses on minwise hashing for binary data. Various extensions to real-valued data are possible. For example, our results naturally apply to *consistent weighted sampling* [25, 15], which is one way to handle non-binary sparse data. The problem, however, is not solved if we are interested in similarities such as (normalized) $k$-way inner products, although the line of work on *Conditional Random Sampling (CRS)* [19, 18] may be promising. CRS works on non-binary sparse data by storing a bottom subset of nonzero entries after applying one permutation to (real-valued) sparse data matrix. CRS performs very well for certain applications but it does not work in our context because the bottom (nonzero) subsets are not properly aligned.

***Building hash tables by directly using bits from minwise hashing***. This will be a different approach from the way how the hash tables are constructed in this paper. For example, [26] directly used the bits from $b$-bit minwise hashing [20, 23] to build hash tables and demonstrated the significant advantages compared to *sim-hash* [8, 12] and *spectral hashing* [29]. It would be interesting to see the performance of this approach in $k$-way similarity search.

$k$-***Way sign random projections***. It would be very useful to develop theory for $k$-way sign random projections. For usual (real-valued) random projections, it is known that the volume (which is related to the determinant) is approximately preserved [24, 17]. We speculate that the collision probability of $k$-way sign random projections might be also a (monotonic) function of the determinant.

## 9   Conclusions

We formulate a new framework for $k$-way similarity search and obtain fast algorithms in the case of $k$-way resemblance with provable worst-case approximation guarantees. We show some applications of $k$-way resemblance search in practice and demonstrate the advantages over traditional search. Our analysis involves the idea of probabilistic hashing and extends the well-known LSH family beyond the pairwise case. We believe the idea of probabilistic hashing still has a long way to go.

## Acknowledgement

The work is supported by NSF-III-1360971, NSF-Bigdata-1419210, ONR-N00014-13-1-0764, and AFOSR-FA9550-13-1-0137. Ping Li thanks Kenneth Church for introducing *Google Sets* to him in the summer of 2004 at Microsoft Research.

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
