[Reviews · NeurIPS 2013]

Submitted by Assigned_Reviewer_4

This paper describes a nice trick to extend MinHash to estimate the generalization of the Jaccard index to more than two sets. Using the trick, the paper presents efficient search algorithms for (1) given S and T, indentifying U for which (S,T,U) are similar, (2) given S, identifying T and U for which (S,T,U) are similar, and (3) identifying S, T and I for which (S,T,U) are similar. There are experiments showing the value of a >2-way similaruty measure.

This paper is not written well and is consequently difficult to read.

As far as I know, the work is original. The algorithms are significant and may prove to be useful in many domains. If this paper is edited to remove all of the grammar errors, it will be very nice.

The anecdotes for the "Google Sets" problem are very cool. I found the search-quality experiments very hard to follow. It would be help a lot if the metric discussion is promoted to come earlier in the section and discussed in a little more depth.

Example presentation issues (restricting to one per page)

Page 1: "one of the most widely used notion" => "notions"
Page 2: "may be" => "maybe"
Page 3: "any two point" => "any two points"
Page 4: "One of the peculiarity" => "peculiarities"
Page 5: "c-BP" in the Section 5 title should be "c-CP"
Page 6: "if we are interested in very high similarity cluster" => "a very" or "clusters"
Page 7: "report top 5 words" => "report the top 5 words"
Page 8: "We formulated ... and obtain" => "formulate" or "obtained"
Summary: I think this is a nice contribution that could have practical impact. It needs a serious edit by a native English speaker.

Submitted by Assigned_Reviewer_5

This paper extends the min-hash mechanism to k-way Jaccard similarity and presents a theoretical analysis following the framework of [Indyk & Motwani '98; Har-Peled et al. '12]. The work is in general solid and supported mostly by the theoretical analysis. Experiments demonstrate the advantage of k-way similarity over the typically pairwise methods.

Overall, I like this paper. Although the majority of the theorems and proofs are based on existing results, the authors have some interesting constructions (e.g., the f_1 and f_2 functions) and covered several aspects of the 3-way scenarios, including near neighbor, close pair and best cluster. One minor concern is that although the arguments of using k-way similarity is convincing, it practical value may not be demonstrated strong enough in the experiments. For instance, the Google set experiment is interesting, but instead of "Sum Resemblance", perhaps the correct function is something that can simulate the conjunction (i.e., w1 is similar to w AND w2 is similar to w). Similarly, in Table 2, the results would be stronger if the pairwise setting became a similar conjunction or average setting. Nevertheless, even if the enhanced pairwise similarity approach performs comparably to the k-way similarity method, the latter is still a more principled approach.

---

Thanks for your response to the comments.
Summary: The proposed approach is a nice extension of min-hash similarity to general k-way Jaccard similarity with a pretty complete theoretic analysis. The experiments could be stronger to better justify the value of k-way similarity over the pairwise case.

Submitted by Assigned_Reviewer_6

This paper addresses the problem of k-way Jaccard-based similarity search. To do so, the authors show that minwise hashing is efficient to address the problem of k-way c-NN search. The development they propose is a direct extension of the one known for pairwise similarities. The extension to other efficient similarities through the PGF transformation is also a direct extension of prior work.

The work presented here is well conducted, clear and of good quality. At the same time, the focus on the Jaccard similarity for sets limits the applicability and the impact of the method. It is not common at all to use presence/absence representations in text mining applications (as the Google set and webspam problems) and the work would be more interesting if different similarities were considered. Regarding the experiments performed, for the Google set problem, I did not find the words obtained by the 3-way similarity really better than the ones obtained by the standard Google algorithm. For the retrieval experiments, it seems obvious that the k-way (k > 2) methods will perform better than the pairwise one, as they use more information (x_1 and x_2, sometimes x_3 vs x_1 only). This experiment is not really convincing and does not show the utility of k-way similarity as compared to say an average of pairwise similarities.
Summary: If the work is interesting, it represents a direct extension of previous studies and is limited in its scope.
Author Feedback

Author rebuttal: Dear Reviewers. Thank you all for the efforts and encouraging comments. Your constructive inputs will be carefully considered.


Assigned_Reviewer_5:

Q:-> The Google set experiment is interesting, but instead of "Sum Resemblance", perhaps the correct function is something that can simulate the conjunction (i.e., w1 is similar to w AND w2 is similar to w).
>> We appreciate reviewers concern here, and in fact it is a very interesting question to find a function which simulates the notion of “AND” well.

If we are constrained with notion of similarity being pairwise (which is the case with current literature) there is not much we can do. The most intuitive way is either to combine the similarity using addition ("Sum Resemblance" in the paper) or use pairwise retrieval for each w_1 and w_2 and do the intersection of top results (PI method in the paper)

Let us say we have found such “ideal” objective which combines similarity between w1 and w2 in the best possible way, we must also ensure that searching with such objective should be efficient. We all agree that efficiency is very crucial in search. Even linear algorithms are prohibitive with the size of web. It should be noted that there is no known provable sub-linear algorithm even for a very simple looking “Sum Resemblance".

3-way resemblance is a very intuitive higher orders similarity, which seems to models the notion of “AND” better than other intuitive pairwise approaches and at the same time it is sub-linear to search.


Assigned_Reviewer_6:

Q:->I did not find the words obtained by the 3-way similarity really better than the ones obtained by the standard Google algorithm.
>> Our aim was to demonstrate the applicability of 3-way similarity for the Google sets application over other similarity measures and not to beat Google’s original algorithm which most likely is much complicated and uses lot more data than 1.2M wiki documents. Interestingly, for the query “finger” and “lakes” we do retrieve better words (see line 354) than Google which is very encouraging. Even for the query “Jaguar” and “Tiger”, 3-way retrieves all the cat family in the top 5 while Google retrieve “dog” as the 5th best candidate which is not a cat family.

Q:->For the retrieval experiments, it seems obvious that the k-way (k > 2) methods will perform better than the pairwise one, as they use more information (x_1 and x_2, sometimes x_3 vs x_1 only).
>> We agree with the reviewer that if we are using more information compared to the pairwise case then we should do better. The experiments establish that 3-way and higher order resemblance search is utilizing this extra information in the right way which we believe is the first step.

The most exciting part which comes from the theory in the paper is that 3-way similarity search admits sub-linear algorithms with same guarantees as the pairwise case. Thus, it is possible to use the extra information in multiple queries, which cannot be used by vanilla pairwise retrieval, without sacrificing computation which is very critical in the big data world.

The biggest disadvantage about sum of pairwise similarity retrieval or any other ways of combining the objective is that it is not known to be efficient. There is no known provable sub-linear algorithm for such combination of similarity.


**************************************

There have been plenty of studies on pairwise similarities but the notion of 3-way and k-way similarities is severely under exploited. We hope our paper will be a good starting point, for an exciting line of research and applications.

We expect many opportunities can be explored. For example, as Reviewer_6 pointed out, it would be useful to test k-way similarity in non-binary (0/1) datasets. We fully agree this would be a promising direction, while here we would like explain this issue a bit more, for the sake of discussions:

1. Other than min-hash, it is difficult to find another good 3-way or k-way similarity estimation algorithms. For example, random projection is extremely popular, but after random projection, all data points become (zero-mean) normal whose distribution is determined by covariance (i.e., pairwise similarity) only.

2. While many datasets used in academia research are not too high-dimensional (e.g., less than 10000 or million) and often not binary, it looks industry indeed uses extremely high-dimensional representations using n-grams. For example, around 2000, industry papers described the use of 7-gram. These days, some search engine applications start to use even 11-grams (by combining all 1, 2, 3, .., 11-grams). With such a high-order n-grams, usually only absence/presence information matters (in fact, most n-grams will occur at most once).

3. On a limited number of datasets we experimented in the past, it looks when the dimensionality of the data is not too high, often the non-binary feature values help (for classification/clustering). But if we use higher n-grams, often only presence/absence matter. For example, using 1-gram (character n-gram), Webspam dataset has 254 dimensions (size of ascii table-1). The classification accuracy using linear SVM is about 93% vs 88% if we use non-binary(tfid) vs binary-only features. With 3-grams, (dimensionality 16 Million) the classification accuracy is 99.6% whether or not we use binary or non-binary (tfid) features. We observed similar phenomenon with other text datasets.

4. Despite the success of using extremely high-dimensional 0/1 high-order n-grams representations, we agree with reviewer that it is nevertheless an interesting research problem to look into non-binary features. The definition of 3-way minwise hashing can be extended to e.g., sum_i min{x_i, y_i, z_i}/max{x_i, y_i, z_i}, which might be something interesting to study.

Thanks again for all your encouraging comments.